# How Does Approaching a Lead Vehicle and Monitoring Request Affect Drivers’ Takeover Performance? A Simulated Driving Study with Functional MRI

**DOI:** 10.3390/ijerph19010412

**Published:** 2021-12-31

**Authors:** Chimou Li, Xiaonan Li, Ming Lv, Feng Chen, Xiaoxiang Ma, Lin Zhang

**Affiliations:** 1CCCC Wenshan Highway Construction & Development Co., Ltd., Wenshan 663000, China; lichimou@outlook.com (C.L.); lvming93@outlook.com (M.L.); 2The Key Laboratory of Road and Traffic Engineering, Ministry of Education, Tongji University, 4800 Cao’an Road, Jiading, Shanghai 201804, China; fengchen@tongji.edu.cn; 3School of Transportation and Logistics Southwest Jiaotong University, Chengdu 611756, China; xiaoxiang.ma@swjtu.edu.cn; 4Shanghai Municipal Engineering Design Institute (Group) Co., Ltd., Shanghai 200437, China; zhanglin_tos@163.com

**Keywords:** conditional automated driving, takeover, functional magnetic resonance imaging, visual behavior, brain activation

## Abstract

With the popularization and application of conditionally automated driving systems, takeover requirements are becoming more and more frequent, and the subsequent takeover safety problems have attracted attention. The present study used functional magnetic resonance imaging (fMRI) technology, combined with driving simulation experiments, to study in depth the effects of critical degree and monitor request (MR) 30 s in advance on drivers’ visual behavior, takeover performance and brain activation. Results showed that MR can effectively improve the driver’s visual and takeover performance, including visual reaction times, fixation frequency and duration, takeover time, and takeover mode. The length of the reserved safety distance can significantly affect the distribution of longitudinal acceleration. Critical or non-critical takeover has a significant impact on the change of pupil diameter and the standard deviation of lateral displacement. Five brain regions, including the middle occipital gyrus (MOG), fusiform gyrus (FG), middle temporal gyrus (MTG), precuneus and precentral, are activated under the stimulation of a critical takeover scenario, and are related to cognitive behaviors such as visual cognition, distance perception, memory search and movement association.

## 1. Introduction

In recent years, automated vehicles have been developed at a fast pace with promising potential to improve road safety, mobility, and provide environmental benefits [1]. According to the taxonomy provided by the SAE International (Society of Automotive Engineers) in 2018, vehicle automation is divided into six incremental levels from Level 0 to Level 5 (SAE standard J3016). Level 0 refers to the full-time performance by the human driver of all aspects of the dynamic driving task, which means all driving operations are manual. The landmark achievement of Level 3 and above is that the automated driving system replaces the manual management of the driving environment. At present, SAE Level 3 and above automatic driving technology is in the stage of development and testing. A Level 3 vehicle is capable of self-control, both longitudinally and laterally; making tactical decisions such as overtaking; monitoring the environment, and alerting the driver if human intervention is required. Whenever a Level 3 vehicle senses a malfunction or encounters a situation beyond the limits of its operational design domain, it issues a Monitoring Request (MR), and then the driver will be required to regain manual control. The requirement to be ready and respond to “a request to intervene in a timely manner” is problematic in terms of human factors and human capabilities [2]. Therefore, research on the driving safety of SAE Level 3 automatic driving technology can help developers and technicians preferably complete the transition from manual to intelligent driving mode.

In SAE Level 3 autonomous vehicles, before being asked to take over, drivers are likely to be engaged in secondary tasks unrelated to driving, such as playing games, watching movies, or attending video conferences. Drivers’ attention that should have been focused on driving is distracted by secondary tasks, which may lead to poor takeover behavior. Merat et al. [3] suggested it could take approximately 40 s for drivers to resume an adequate and stable lateral control of driving when they switch from autonomous to manual driving. Similarly, Pampel et al. [4] found a diminishing difference in mean speed, measured using 5-s time bins, after approximately 20 s of vehicle control take-over and standard deviation of lane position did not stabilize for the first 10 s during a non-distractive short take-over driving scenario. Vogelpohl et al. [5] argued that drivers might require additional time and assistance in order to reach a level of situational awareness necessary to resume manual driving. Many other literatures have studied the ability of human drivers to take over control from the automated vehicle for various types of traffic situations [6,7,8,9,10]. Several bold simulator studies have found that in urgent conditions, mean take-over time of drivers is even as low as about 1 s [11,12,13]. In a meta-analysis of 129 experimental studies, Zhang et al. [14] examined the determinants of take-over time and found that take-over times depend on, among other factors, the urgency of the situation and the type of take-over request. Based on the above research, the speed difference between the front and the lead vehicles and the takeover monitoring request are set as the experimental variables.

In recent years, the number of studies on brain activity in the conversion process of conditional automatic driving is also increasing. Tsunashima and Yanagisawa [15] used functional near-infrared spectroscopy (fNIRS) to evaluate the brain function of car drivers with and without Adaptive Cruise Control (ACC) systems and demonstrated that the frontal lobe was less active during ACC drive. Cao et al. [16] showed how electroencephalogram (EEG) signal can be used as a measure of driver’s attention to random takeover tasks. Arakawa et al. [17] studied the psychological assessment of driver mental state in autonomous vehicles using fNIRS signals in combination with other biomarkers such as blood pressure, body pressure distribution, salivary monitoring and eye tracking. Lee and Yang [18] tested a range of takeover transition alerts while analyzing drivers’ EEG brainwaves. The transition alert system that used a combination of auditory, visual and haptic stimuli proved to be most effective. Due to the complexity of the fMRI instrument and the particularity of the experimental design, there are few results from research using fMRI technology to study the takeover process. Compared with EEG, fMRI has higher spatial resolution and good temporal resolution. Therefore, fMRI can dynamically analyze the activation changes in the spatial structure of brain regions, which cannot be done by other methods. Compared with static MRI, fMRI presents dynamic profile on this basis. Critical degrees was set as the only variable, so as to improve the rationality of the experiment and the reliability of the conclusion.

Therefore, functional magnetic resonance imaging (fMRI) technology combined with the driving simulation experiment was applied to study in depth the effects of critical degree and monitor request (MR) 30 s in advance on drivers’ visual behavior, takeover performance, and brain activation.

## 2. Driving Simulation Experiment

### 2.1. Driving Simulator

The driving simulation equipment combination was composed of a steering wheel, driving simulation seat, and three TV screens, which provided a comprehensive and scientific technical support for the design of the experiment and the analysis of subsequent data (Figure 1). These three screens could provide about 135° field of vision, creating a more realistic and three-dimensional driving experience. The rear-view mirror and instrument panel were also included. The simulation software platform can record more than 400 experimental parameters at the same time, including vehicle motion parameters, driver operation parameters, road environment parameters, traffic parameters, etc.

Eye tracker equipment was used to record participants’ eye movement. Tobii Pro is a wearable eye tracker developed by Tobii company of Sweden (Tobii Pro China office, Shanghai, China). It has the function of real-time observation and video recording, and is suitable for research in various driving environments. The head weight of the eye tracker is only 45 g, which ensures the comfort and head freedom of the experimenters. In addition, the eye tracker is equipped with two eyes, a four lens eye tracker, and an ultra wide-angle camera, which can accurately capture the eye movement behavior of the experimenters and the changes of the areas concerned.

### 2.2. Driving Scenario and Experiment Design

Highways have clear road signs and road markings, a closed road environment and relatively stable traffic flow. Compared with complex urban roads, highway roads have less interferences and are less dangerous. Therefore, highways have become one of the main public places for automobile corporations and Internet enterprises to test unmanned vehicles. However, when a self-driving vehicle pulls out of a highway, the driver is required to take over the vehicle and faces a more complex traffic environment on the city’s roads. Therefore, a highway ramp was selected as the driving scenario: when the self-driving vehicle arrives in the deceleration lane, it is then controlled by the driver and driven off of the highway at high speed.

The experimental scene was set as a two-way six-lane highway with a continuous emergency lane on the right side, and the width of each lane was 3.5 m. The highway design speed was set as 100 km/h and the ramp design speed was 50 km/h. The parameters set for autopilot were as follows: the initial speed was 0 km/h, gradually increased the speed to the maximum speed of 100 km/h, and the ego vehicle kept the speed of 100 km/h on the right lane of the highway. The takeover mode was set to press the button on the steering wheel. In order to give participants a different degree of urgency when they took over, a lead vehicle was set up in front of the ego vehicle and a certain distance was maintained according to the needs of each scenario (Table 1). In order to observe the true state of the participant facing the risk of takeover, there would be a risk of collision if the participant refused or neglected to take over the vehicle.

According to the Chinese code GB5768-2009 for exits, exit notice signs are, respectively, set at 2 km, 1 km, 500 m and at the starting point of the transition section from the deceleration lane of expressway or urban expressway. In this experiment, the ego vehicle speed was set to 80 km/h. According to Chinese regulations, the last exit sign (the sign at 500 m) was selected as the place where MR was triggered. After calculation, it needed to be at least 23 s in advance. When added to the time to collision at 7 s, it was finally determined that MR was set as 30 s in advance.

Finally, six experimental scenarios were designed. The differences between them lied in the speed difference between the ego vehicle and the lead vehicle when taking over (0 km/h or 30 km/h), the number of seconds before taking over (5 s or 7 s), and whether to give a monitoring request (MR) 30 s before taking over (yes or no). Before each takeover, there was an automatic driving process that last 210 s (3.5 min), in order to immerse participants in mobile gaming tasks during this period.

### 2.3. Participants and Experiment Procedure

Thirty-nine participants (26 males and 13 females) between the ages of 20 to 56 years (M = 29.8, SD = 10.4) participated in the experiment. All participants had valid driving licenses for at least two years (M = 6.2, SD = 4.6), normal or corrected to normal vision and normal color vision. During the experiment, the participants were in a good mental state, without major medical history, physiological or psychological diseases, or bad habits that might affect the experimental results.

After filling in the personal information form, participants read written instructions about the simulated vehicle, the surrounding devices, and the experimental procedure. Then participants put on the eye tracker and set the initial parameter values. To ensure the smooth progress of the experiment, participants were again told when and how to take over the vehicle and placed in a dedicated test scenario to be familiar with the equipment and takeover methods. In the formal experiment, the initial state was that the test vehicle was in the rightmost lane and automatic driving state. Participants were required to invest in the mobile game until they heard the takeover command, then they needed to press the button on the steering wheel to take over the ego vehicle and drive it out of the highway ramp. Participants were assigned to six driving scenarios in random order.

### 2.4. Data Collection and Analysis

The output frequency of driving simulation software was 10 Hz, and that of the Tobii eye tracker was 50 Hz. The collected data mainly included four aspects: driver operation, vehicle trajectory, vehicle running state, and driver visual behavior.

When the angular velocity of the visual angle detected by the eye tracker was less than 30 degrees per second, it was considered as fixation behavior, while when the angular velocity was higher than 30 degrees per second, it was considered as saccade behavior. The threshold of fixation time was 60 ms. If the time interval between two fixation points was less than 20 ms, and the angle difference between them was less than 0.5 degrees, they were combined into one fixation point. In addition, the areas of interest (AOIs) of the test video were drawn frame by frame, including the front road area, the instrument panel area, the left and right rear-view mirror area, and the central rear-view mirror area (Figure 2).

## 3. fMRI Experiment

In order to further explore the differences of brain activity before and after taking over the vehicle, a functional magnetic resonance imaging (fMRI) instrument was used to explore the brain activity in different critical scenes. “Block design” is the most common rule of designing fMRI experiments. Each stimulus is called a “block”, and all blocks appear in turn with the resting state. The fewer types of stimuli, the more cycles of repeated stimuli, and the more accurate the results. The variables of driving simulation experiment involved hearing (monitoring request, MR) and vision (critical or not). Considering the high noise in the nuclear magnetic instrument, only the visual variables were retained. The brain load of drivers was analyzed to explain the neural activity when the driver suddenly transfers from secondary task to driving task.

### 3.1. Apparatus

The fMRI instrument’s imaging data drift rate is as low as 0.03%, so it can conduct more accurate and scientific brain functional imaging (Figure 3). The whole brain scanning time (TR) of the instrument is 2 s, 40 slices are scanned, and the slice thickness is 3 mm. The scanning sequence is from bottom to top, and the default voxel size is 3 mm × 3 mm × 3 mm.

In order to enable the participant to receive the visual stimulation of the driving scene in the fMRI instrument, the projection board was suspended above the eyes, and the image was projected onto the board through mirror reflection so that the video image appeared in front of the participant’s line of sight. There were also two keyboards with four buttons in total, which were put in the hands of the participant for judgment, selection, and other behaviors in this experiment (Figure 4).

### 3.2. Driving Scenario and Experiment Design

One of the main experimental design ideas in fMRI research is called subtraction strategy, also called block design. Its principle is that the subjects receive continuous stimulation for a period of time. There are two kinds of stimulation tasks, one is the stimulus task with the research purpose such as judgment and thinking, and the other is not included, which is called the control task. Each group can be alternately or randomly appeared with the same duration of stimulation. The block experiment design is characterized by a block form showing a period of stimulation, which can be compared with the changes of brain oxygen content produced by two groups of block stimulation to obtain the functional localization of brain structure activity related to task stimulation.

In this experiment, two scenarios with different degrees of criticality in the process of taking over were selected for research and driving simulation platform was used to record 10 s driving video to present stimulation. The biggest difference between the two scenarios was that the first scenario was non-critical takeover, and the distance between the ego vehicle and the lead vehicle remained unchanged (consistent with the Non-critical without MR scenario in driving simulation experiment). The second scenario was critical takeover, and the distance between the ego vehicle and the lead vehicle gradually shortened (consistent with the Most-critical without MR scenario in driving simulation experiment) (Table 2 and Figure 5).

In the fMRI experiment, before the visual stimulation, common sense questions were set to simulate the secondary task state during automatic driving. Participants needed to answer the questions by pressing the keyboard. After a period of time, the text “please take over” appeared on the screen, asking the participant to take over the vehicle through the keyboard of the other hand. At this time, the projection version started to present driving video stimulation (Figure 6).

In order to reduce the effect of head-shaking on the imaging quality, the scanning time should not be too long. Therefore, the scanning was divided into three time periods. After each scanning period, the participant could rest for a few minutes. The scanning sequence and time are shown in Table 3. Critical stimulus and non-critical stimulus were randomly presented by E-Prime software. A total of 18 stimulus tasks (nine times each) were presented in three scans.

### 3.3. Participants and Experiment Procedure

This experiment required that the subjects did not have any mental illness or nerve/brain related injury. In addition, the participants were required to have no claustrophobia, pacemaker, stent or other metal embedded in their bodies. All participants were forbidden to smoke, drink alcohol, drink coffee or any external behavior that may have affected their brain activity within 24 h before the experiment. They were asked to sign the “informed consent form” and “safety screening form” according to the regulations of the school’s laboratory.

Although fMRI is not radioactive and is harmless to the human body, not everyone can accept that their brains are put into this rumbling behemoth. Therefore, out of a voluntary principle, we replaced and added some experimenters, considering the balance of the sample. A total of 30 participants were selected for the fMRI experiment, including 19 males and 11 females. The participants ranged from 20 to 31 years old (M = 24.1, SD = 2.7), and all participants had more than two years of driving experience. Their vision was corrected to 0.8 through the non-magnetic glasses equipped in the laboratory, and they were in a good mental state during the experiment.

After being informed of the experimental rules and signing the form, all participants passed the metal screening test or changed into metal free pajamas if necessary, and wore metal free glasses and earplugs. We then let them lie down in the fMRI instrument and hold one keyboard in each hand. Their heads were fixed with a sponge and they were told that they could exit the experiment at any time by pressing the ballonet. The positioning scanning was performed before the three scans of the formal experiment, and there was also a five-minute structural scanning afterwards, which belonged to the necessary scanning process of all fMRI experiments. The whole scanning process lasted about half an hour.

### 3.4. Data Collection and Analysis

In this experiment, the scanning parameters were set as follows: repetition time (TR) = 2 s, tilt angle = 90 degrees, slice thickness = 3 mm, slice layer number = 40, scanning order was bottom-up, the size of each voxel was 3 mm × 3 mm × 3 mm.

In order to standardize each participant’s brain and correct the error caused by shaking their heads during the experiment, the fMRI image was preprocessed spatially, including slice time correction, motion correction, normalization, smoothing, etc.

After pre-processing the fMRI data of 30 participants, the matrix based on the general linear model was established in the software package SPM (Statistical Parametric Mapping). The linear model proposed by Friston et al. is currently one of the most internationally recognized fMRI data processing methods [19]. Its regular form can be written as:(1)Y=βX+ε
where Y is the observed data matrix, the columns of the matrix represent the total number of voxels, and the rows are the number of scans. X is the regression factors matrix, the number of rows is the number of scans, and the number of columns is the number of factors. Usually, the first column is the stimulation task (critical), the second column is the micro stimulation task (non-critical), and the remaining columns are simulated noise, including data such as head movement. β is an unknown parameter matrix, each representing an unknown parameter vector of each voxel. ε is the random error vector.

In this experiment, the main factors X affecting the brain nerve activity were selected as follows: critical takeover stimulation, non-critical takeover stimulation, three direction head movement factors (two-dimensional displacement “head movement1” and “head movement 2” and rotation “head movement 3”). The formula is converted to:(2)$$\text{Y}=\beta_{0}+\beta_{\text{critical}}\cdot {\text{X}}_{\text{critical}}+\beta_{\text{non}-\text{critical}}\cdot {\text{X}}_{\text{non}-\text{critical}}+\beta_{\text{head}\;\text{movement}\;1}\cdot {\text{X}}_{\text{head}\;\text{movement}\;1}+\beta_{\text{head}\;\text{movement}\;2}\;\cdot {\text{X}}_{\text{head}\;\text{movement}\;2}+\beta_{\text{head}\;\text{movement}\;3}\cdot {\text{X}}_{\text{head}\;\text{movement}\;3}+\varepsilon$$

After pretreatment, the number of voxels in the standard brain was 52 × 62 × 53, that was, about 170,000 voxels. Set *p* value to 0.001 when activating *t* test for each voxel. Only when continuous voxels are activated, it is considered that a certain brain region is significantly activated. The resulting brain activation can be graphically represented by color coding, that is, the activation intensity of different regions of the brain.

In order to distinguish the extra effects of a critical takeover scenario on the driver’s brain activity, the activation and deactivation results were used to represent the brain activity. The activation was the subtraction of non-critical takeover stimulus from critical takeover stimulus. Deactivation was opposite to activation, which was to subtract critical takeover stimulus from non-critical takeover stimulus.

## 4. Results and Discussion

In driving simulation experiment, drivers’ visual behavior and take-over performance were analyzed in six different scenarios: non-critical, most-critical, sub-critical, non-critical with MR, most-critical with MR and sub-critical with MR. In the fMRI experiment, the fMRI instrument was used to further analyze the brain activation during driver’s takeover in critical and non-critical scenarios.

### 4.1. Driving Simulation Experiment

#### 4.1.1. Visual Behavior

Visual reaction time refers to the time from the appearance of the alert prompt to the time the participant looked at the road environment, which can reflect the speed of drivers’ visual attention shift (Table 4 and Figure 7). Results showed that the main factor affecting drivers’ visual reaction time was whether or not the monitoring request (MR) 30 s before takeover was expected. In the absence of MR, drivers’ average visual reaction time was around 2 s, mostly between 1.4–2.7 s. MR can significantly shorten the visual reaction time and reduce its dispersion, as in the last three scenarios, drivers’ visual reaction time was mostly in 0–0.5 s.

For the viewpoint thermal map, the greater the brightness of a specific point and the redder the color, the more attention the driver pays to the point. It can be seen that the participants’ viewpoints were mainly focused on the lead vehicle and the dashboard (Figure 8). For scenes without MR, participants paid more attention to the takeover button on the steering wheel, which may cause driving distraction. The MR can divert part of the attention on the takeover button to the lead vehicle and promote participants to better observe the situation ahead.

It was found that the total fixation duration of the front road area increased in all scenarios with MR compared to those without MR. This meant that since participants had fully observed the surrounding driving conditions during the pre-alert period, they could more calmly allocate more time to the road ahead after taking over, rather than observing everywhere aimlessly and in a panic (Table 5). Because they were more leisurely, participants in the scenarios with MR looked at the road ahead more often than those in the scenarios without MR, and because of the increase of points, the average single fixation time in some scenarios with MR decreased slightly (Figure 9).

The change of pupil diameter can reflect the intensity of ambient light, driver’s tension, cognitive load, and fatigue. In addition, the light in the laboratory was set at “constant”. Every time the participant took over the vehicle, his/her viewpoint was transferred from the same mobile phone screen to the simulator screen. In the subsequent horizontal comparison of each scenario, the influence of light factor on the results was very limited. Therefore, pupil diameter was used to express the driver’s visual cognitive load and tension. Normally, the pupil diameter of a person is between 2 and 6 mm. In special cases, the pupil diameter ranges from 1.5 to 8 mm. Figure 10 shows the change curve of pupil diameter of a driver in the most-critical scenario without MR before and after the taking over prompt. During the automatic driving, the pupil diameter of the driver fluctuated between 3.5–4.0 mm when being engaged in the secondary task. When the takeover command sounded, the pupil diameter increased rapidly, and gradually decreased after reaching the maximum of 5 mm.

There were differences in the pupil diameter of the driver’s left and right eyes, but the overall change law was consistent, so the average pupil diameter of both eyes was used in the follow-up analysis. In order to explore whether there were significant differences in the pupil diameter of drivers in different scenarios, a paired sample *t*-test was conducted for the average pupil diameter of both eyes in each scenario (Table 6).

It was found that whether there is MR or not, if there is a speed difference between the ego vehicle and the lead vehicle, the pupil diameter will increase significantly, which means that the visual cognitive load is heavy in critical scenarios. However, the safety distance reserved during taking over has no obvious effect on the pupil size of the driver. In addition, MR can significantly reduce the pupil diameter of the driver when there is a most-critical takeover situation (Scene 2, 5, *p* = 0.011), and reduce the driver’s tension and mental load in the face of danger, so as to improve safety.

#### 4.1.2. Takeover Performance

Takeover time refers to the time from the system issuing takeover request to the driver regaining the driving control of the vehicle. In this experiment, takeover time was recorded from the prompt of takeover request to the participant’s press of the takeover key. It can be seen that the takeover time with MR was generally less than that without MR (Figure 11). The takeover time with MR was between 0.4–3.4 s, while that without MR was between 1.3–4.0 s. The average takeover time of non-critical scenario without MR was the largest.

So, is the way of taking over dangerous? According to the sequence of viewpoint regression times and manual driving start times, take over mode can be divided into two types: (mode A) takeover driving manually after viewpoint regression, at this time, drivers or vehicles monitor and make decisions on the surrounding environment in the whole process of taking-over; (mode B) the viewpoint returns after taking over. There will be a dangerous period when neither the driver nor the vehicle system detects the surrounding environment (Figure 12 and Table 7). In scenarios without MR, that was, the participant was still in the secondary task state when receiving the takeover prompt, about 17.9%~25.6% of the participants took over the vehicle before viewpoint regression, while in scenarios with MR, less than 2.6% of the participants took over in mode B. This means that MR can make the driver’s viewpoint return in advance and earlier than the time of real takeover, so as to improve the safety of takeover.

The longitudinal average velocity can reflect the driver’s ability to control the vehicle speed after takeover and the dynamic performance of the vehicle in the longitudinal direction. Taking 0.5 s as the unit time interval, the average longitudinal speed change trend of drivers in 7 s after taking over the vehicle (starting from pressing the take-over button) was calculated (Figure 13). It can be seen from the figure that the starting speed of Scene 1 and Scene 4 (non-critical scenes) was 50 km/h, and the speed curve was smoother than other critical scenes. The starting speed of critical scenes was 80 km/h, and the braking and acceleration process of Scene 2 (most-critical scene without MR) was the most severe, and its speed fluctuation was the largest. All scenes generally reached the minimum speed within 4.5–5 s after taking over, and then the vehicle speed rose slowly.

For the stage before the vehicle speed reached the minimum value, that is, the average longitudinal speed within 5 s after the driver took over, the box plot was drawn (Figure 14). Non-critical scenes have the most concentrated average speed distribution, while the speed distribution in most-critical scenes is the most dispersed, which indicates the increasing speed difference in a critical scene.

In order to explore whether the reserved safety distance and MR have a significant impact on the speed control, the paired sample *t*-test was conducted for the longitudinal velocity mean of each scene (Table 8). The results show that the reserved safety distance has a significant effect on the vehicle speed after taking over. The shorter the distance, the more intense the driver’s braking reaction and the lower the average speed in 5 s. In addition, the results also showed that the presence or absence of MR has a significant effect on the most-critical take over scene, but has no significant effect on the non-critical and sub- critical take over.

Similarly, take 0.5 s as the unit time interval, average longitudinal acceleration change in 7 s after taking over (starting from pressing the take-over button) was calculated (Figure 15). It can be seen that the acceleration of non-critical scenes was relatively stable. The longitudinal acceleration of other scenes had similar change trends, which could be roughly divided into three stages within 7 s: (1) the absolute value of deceleration increased gradually; (2) the absolute value of decreased gradually; (3) the acceleration or deceleration values tended to be stable. The acceleration change of the most-critical scene without MR was the most dramatic, while the acceleration change of the sub-critical scene with MR was the slowest.

The absolute value of acceleration in the non-critical scenes was small, and the distribution was relatively concentrated (Figure 16). For other critical scenarios, the box chart of the most-critical scene is the longest, which indicates that the drivers’ choice of acceleration was the most dispersed and their driving behavior was not consistent, which may lead to the most complex speed change.

In order to explore whether the reserved safety distance and MR have a significant impact on the absolute value of longitudinal acceleration, the paired sample *t*-test was conducted (Table 9). The results showed that the reserved safety distance has a significant effect on the absolute value of acceleration. When the reserved safety distance increases, the absolute value of acceleration decreases significantly, speed fluctuation becomes smaller, and the acceleration and deceleration behavior is smoother. For those critical scenarios, MR can reduce the average absolute value of acceleration, significantly reduce the sharp change of vehicle speed, and improve driving comfort and safety. However, the effect of MR is not significant for the non-critical scenario.

Lateral offset represents the lateral driving stability, with positive and negative distinction between left and right. The absolute values of the maximum lateral acceleration can also indicate whether there has been violent driving behavior. According to the test data, there was little difference in the lateral offset and the absolute values of the maximum lateral acceleration in the six scenes. *T*-test also shows that critical or not, the size of the reserved safety distance or whether there is MR had no significant effect on the lateral offset.

In addition, through paired sample *t*-test for the standard deviation of lateral displacement in different scenes, it was determined that critical or not has the most significant impact on the standard deviation of lateral displacement, while safety distance and presence of MR have no significant impact (Table 10). The results suggest that reducing the workload or urgency of operation at the initial stage of taking over can significantly reduce the standard deviation of vehicle lateral displacement and improve the driver’s lateral control performance.

### 4.2. fMRI Experiment

The purpose of fMRI data analysis is to detect correlations between brain activation and tasks performed by participants during scanning. It also aims to discover correlations with specific cognitive states, such as memory and recognition.

The analysis was based on the individual average data. The coordinates of brain activation were calibrated with Brodmann area and Anatomical Automatic Labeling (AAL) area, respectively. The number of active voxels, *p* value, coordinate partition and image are shown in the following charts (Table 11 and Figure 17).

The *p* value reflects the significant difference between the critical and non-critical conditions (Table 11). Activation areas with *p* < 0.05 were selected for follow-up analysis.

First, the most significant activation was in the visual cortex: increased activation of the middle occipital gyrus (MOG) and fusiform gyrus (FG) was detected in the participants’ brains under critical takeover. The role of MOG and FG has been clearly defined in medicine, of which the MOG is mainly responsible for object recognition, including visual recognition and perception; the FG is located in the Brodmann area 37 and while its exact function is still controversial, the confirmed functions include color information processing, face and text recognition. These results confirm that the critical takeover brings more visual stimulation to the driver than the non-critical scenario, and requires greater visual observation, which is in line with the expectation.

Secondly, the middle temporal gyrus (MTG) and precuneus were significantly activated in the critical takeover scenario. The known functions of the MTG include distance perception and text recognition; while the precuneus participates in memory tasks, such as when people watch a picture and recall what they remember according to the picture [20]. In addition, it also participates in the content related to visual space, such as when people perform an action, or imagine or prepare to perform the action, as the precuneus will guide the process of spatial attention transfer [21].This shows that critical takeover brings visual stimulation, but also brings greater visual space sense of stimulation. The activation of the MTG and precuneus proved that there were processes of distance perception, memory retrieval (maybe driving experience, emergency response experience, etc.) and spatial attention transfer after the driver took over the vehicle.

What’s more, the precentral gyrus (PG) in the motor cortex related to the control of muscle movement was significantly activated by the critical takeover process. In the fMRI instrument, the participants were required to keep the whole body motionless, so the activation of the PG did not seem to meet the expectation. However, according to previous fMRI studies, many scholars have found that the PG is related to car driving. For example, Bernardi et al. [22] found that compared with novice drivers, experienced drivers activate the PG more significantly during driving. Gallese et al. [23] and Iacoboni et al. [24] have also proved that the activation of PG is also related to action planning, observation, and imagination of motor actions. It can be inferred from the results that the driver plans and associates the action more strongly in the critical takeover scenario, which may include braking, steering, honking, and other actions.

For the sudden takeover request, whether the driver who focused on the secondary tasks can keep a reasonable and safe distance from the car in front, whether he can shift his attention to the driving task in time, and whether he can quickly restore his driving ability are always key issues in the takeover conversion. Therefore, from the perspective of a driver’s brain activity, whether or not people with damaged areas of MTG, precuneus or PG are suitable for driving under automatic driving conditions of L3 and below remains to be further studied, which also provides a research idea for how to evaluate the driver’s ability to take over the vehicle in the future.

According to Table 12, only partial deactivated voxels were found in the precentral, but only 35 of them were deactivated, which did not meet the requirement of significant activation (*p* = 0.144 > 0.05). Therefore, the following conclusion is drawn: compared with critical takeover, non-critical takeover does not lead to additional significant activation of brain areas.

## 5. Conclusions

The present study used functional magnetic resonance imaging (fMRI) technology, combined with driving simulation experiment, to study in depth the effects of critical degree and monitor request (MR) 30 s in advance on drivers’ visual behavior, takeover performance and brain activation.

Results show that MR can effectively improve the driver’s visual and takeover performance, which is consistent with the existing conclusions [12,13,25]. MR can significantly shorten the visual reaction time, reduce driver distraction, and reduce visual pressure in most-critical scenes. MR can divert part of the attention on the takeover button to the lead vehicle and prompt participants to better observe the situation ahead. Moreover, MR can significantly reduce the pupil diameter of the driver when there is a most-critical takeover situation. In terms of takeover performance, MR can effectively reduce the takeover reaction time and make the driver’s viewpoint return earlier than the time of real takeover, so as to improve the safety of takeover. The presence or absence of MR has a significant effect on the most-critical takeover scene but has no significant effect on the non-critical and sub- critical take over. For those critical scenarios, MR can reduce the average absolute value of acceleration, significantly reduce the sharp change of vehicle speed.

Besides, the length of the reserved safety distance can significantly affect the distribution of longitudinal acceleration. Critical or non-critical takeover has a significant impact on the change of pupil diameter and the standard deviation of lateral displacement. This means that criticality significantly affects the driver’s tension and lateral performance, which is consistent with the results of many studies [6,14]. When the reserved safety distance increases, the absolute value of acceleration decreases significantly, the speed fluctuation becomes smaller, and the acceleration and deceleration behavior is smoother. It was also determine that whether there is MR or not, if there is a speed difference between the ego vehicle and the lead vehicle, the pupil diameter will increase significantly, which means that the visual cognitive load is heavy in critical scenarios.

Under the stimulation of critical takeover scenario, five brain regions, including the middle occipital gyrus (MOG), fusiform gyrus (FG), middle temporal gyrus (MTG), precuneus and precentral are activated, which are related to cognitive behaviors such as visual cognition, distance perception, memory search, and movement association. Although the activation of the precentral gyrus (PG) related to movement is not consistent with the static experimental state, according to other fMRI studies, many scholars have found that the activation of the PG area is related to action planning, observation and the imagination of motor actions [22,23,24]. Therefore, it can be inferred that in the critical takeover scenario, drivers’ plans and the relevance of the action are stronger. At the same time, the results show that the non-critical takeover stimulation does not lead to additional significant activation areas compared with the critical takeover stimulation.

Further research will be carried out to improve the fMRI equipment and realize the effect comparison of haptic takeover and visual takeover. In addition, the activation state of brain regions under dynamic fMRI experiments is also worthy of an in-depth study.

## Figures and Tables

**Figure 1 ijerph-19-00412-f001:**
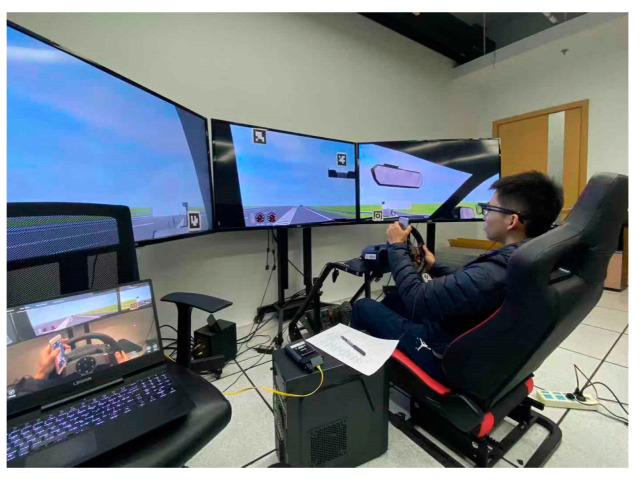
The driving simulation equipment.

**Figure 2 ijerph-19-00412-f002:**
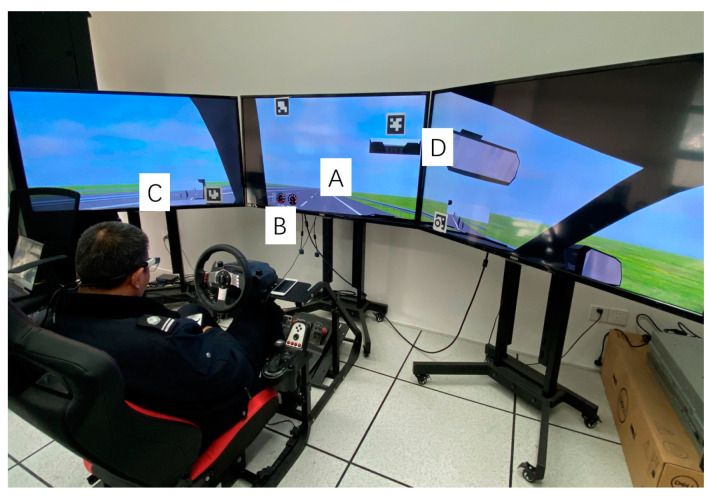
Visual inspection areas include: (**A**) road ahead; (**B**) dashboard; (**C**) left rearview mirror; (**D**) central rearview mirror.

**Figure 3 ijerph-19-00412-f003:**
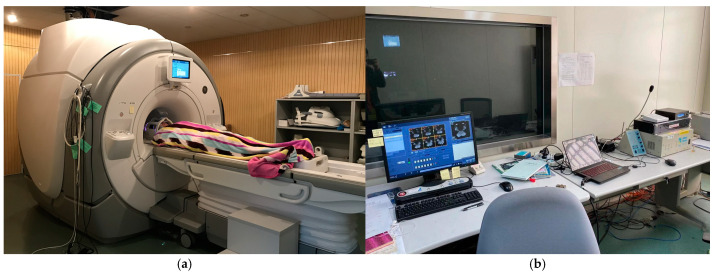
(**a**) fMRI instrument; (**b**) fMRI console.

**Figure 4 ijerph-19-00412-f004:**
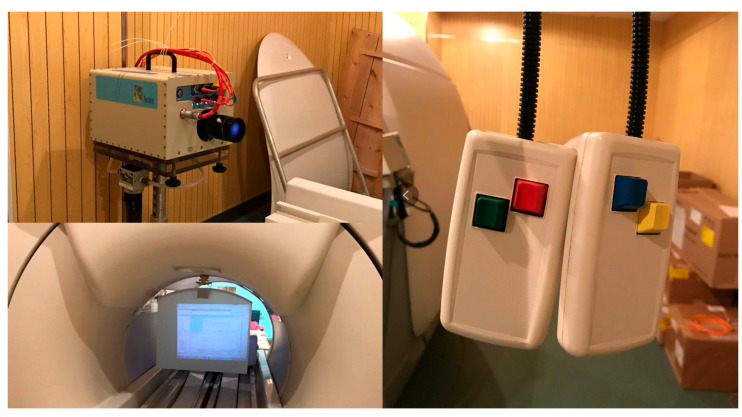
Projector, projection board and key boards.

**Figure 5 ijerph-19-00412-f005:**
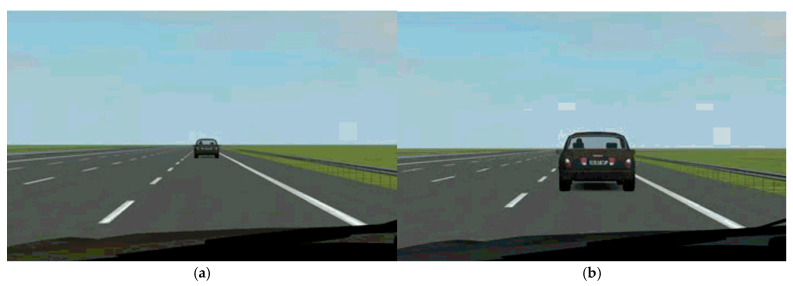
Two different stimulation scenarios in fMRI experiment: (**a**) non-critical takeover; (**b**) critical takeover.

**Figure 6 ijerph-19-00412-f006:**
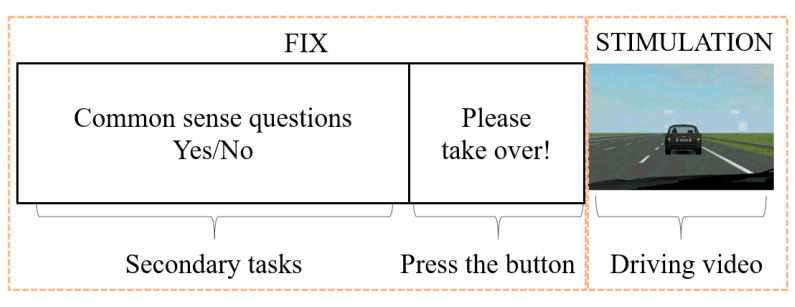
Flow diagram of a complete fMRI experiment of simulated takeover (the scan between adjacent stimulus tasks is called “fix”.).

**Figure 7 ijerph-19-00412-f007:**
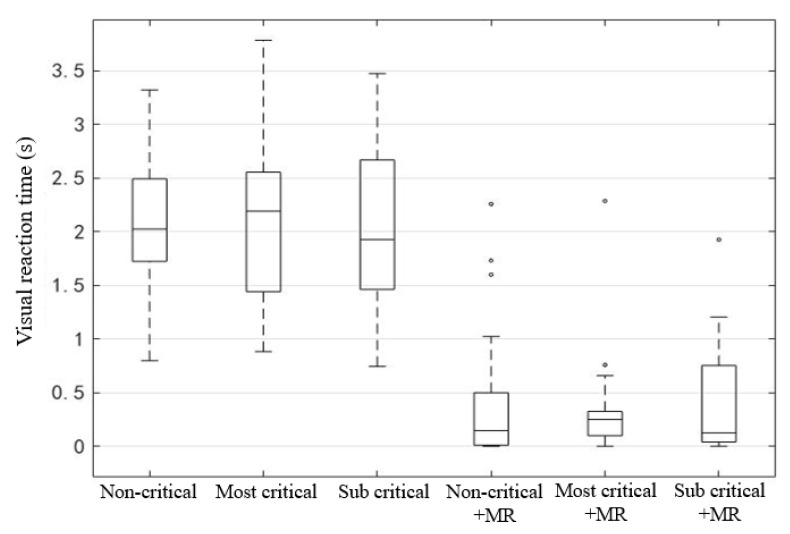
Visual reaction time of six different scenarios.

**Figure 8 ijerph-19-00412-f008:**
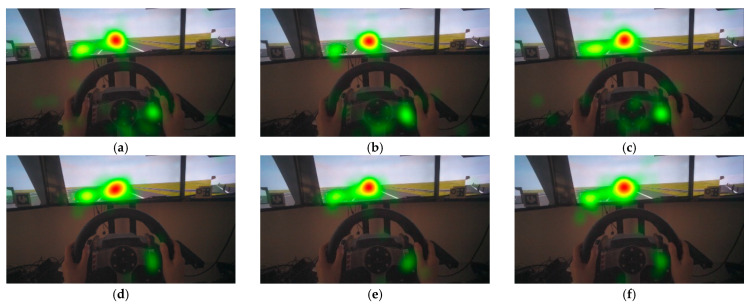
Thermal distribution of viewpoint: (**a**) non-critical; (**b**) most-critical; (**c**) sub-critical; (**d**) non-critical with MR; (**e**) most-critical with MR; (**f**) sub-critical with MR.

**Figure 9 ijerph-19-00412-f009:**
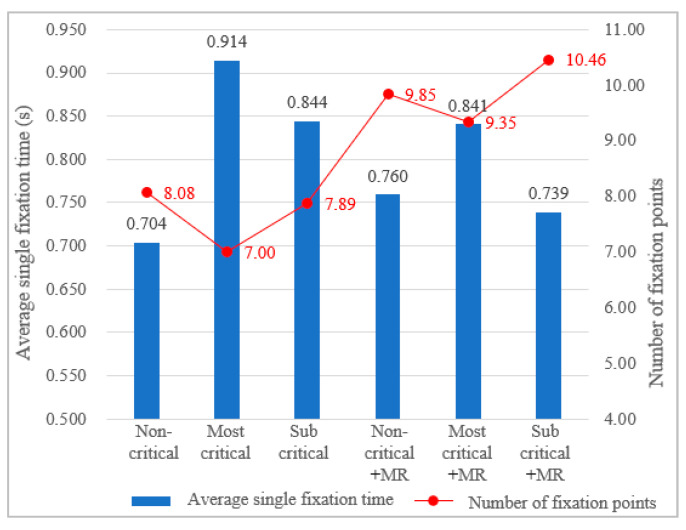
Number of fixation points in the front road area and average single fixation time.

**Figure 10 ijerph-19-00412-f010:**
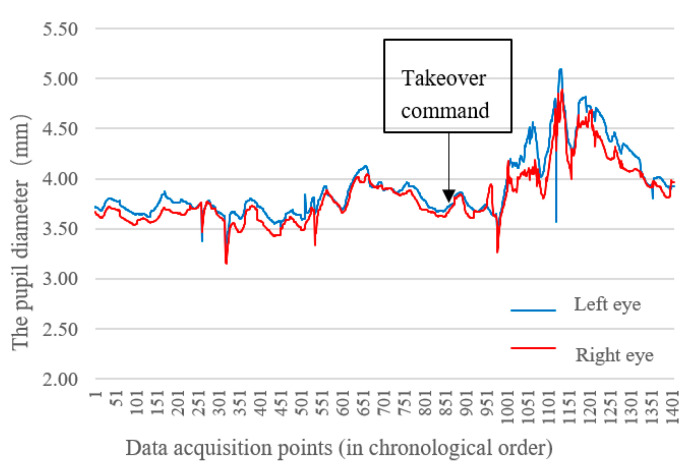
Change of pupil diameter of both eyes during takeover process in the most-critical scenario without MR.

**Figure 11 ijerph-19-00412-f011:**
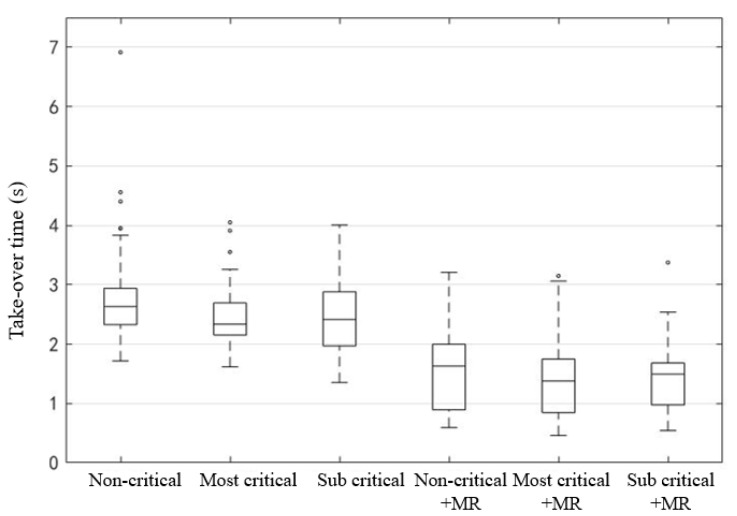
Takeover time of six scenarios.

**Figure 12 ijerph-19-00412-f012:**
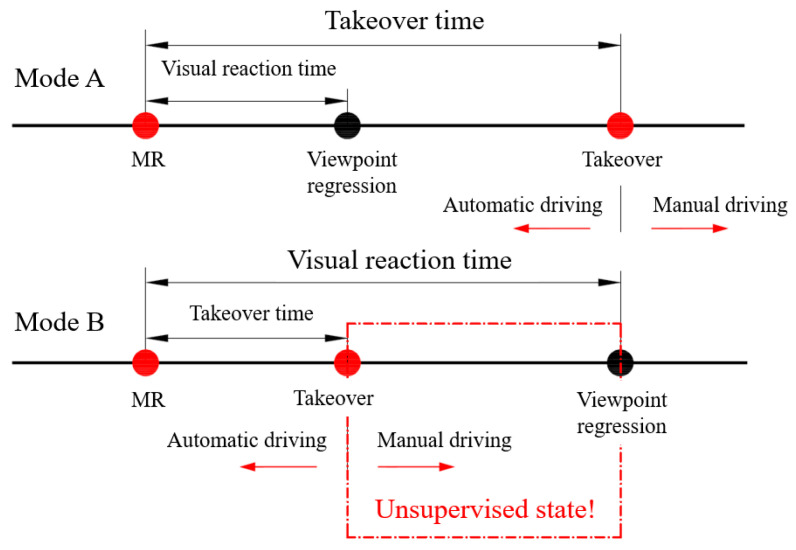
Two modes of takeover.

**Figure 13 ijerph-19-00412-f013:**
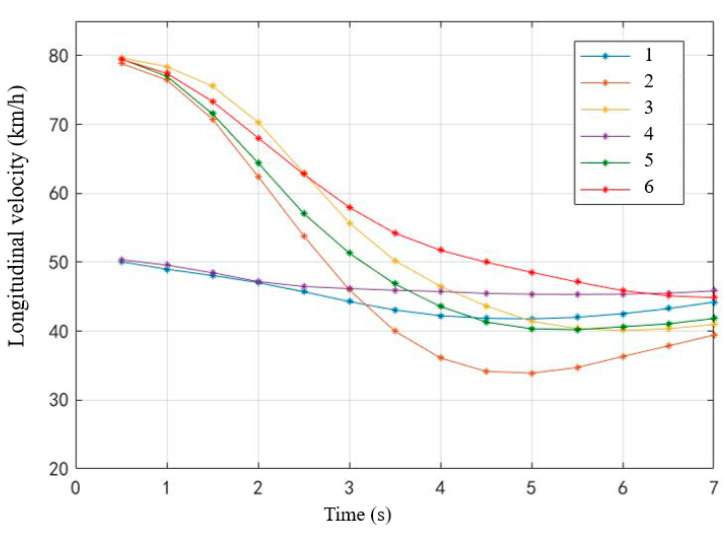
Change curve of mean velocity within 7 s after taking over.

**Figure 14 ijerph-19-00412-f014:**
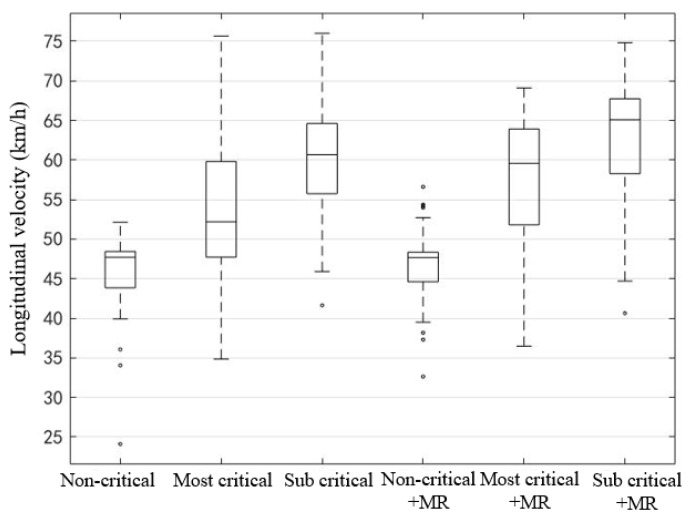
Box plots of mean longitudinal speed within 5 s after taking over.

**Figure 15 ijerph-19-00412-f015:**
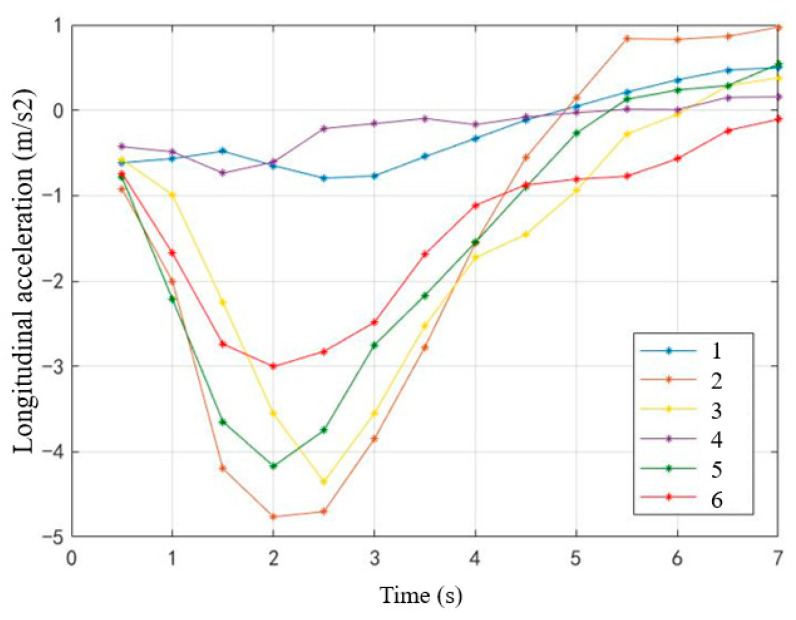
Mean change curve of longitudinal acceleration within 7 s after taking over.

**Figure 16 ijerph-19-00412-f016:**
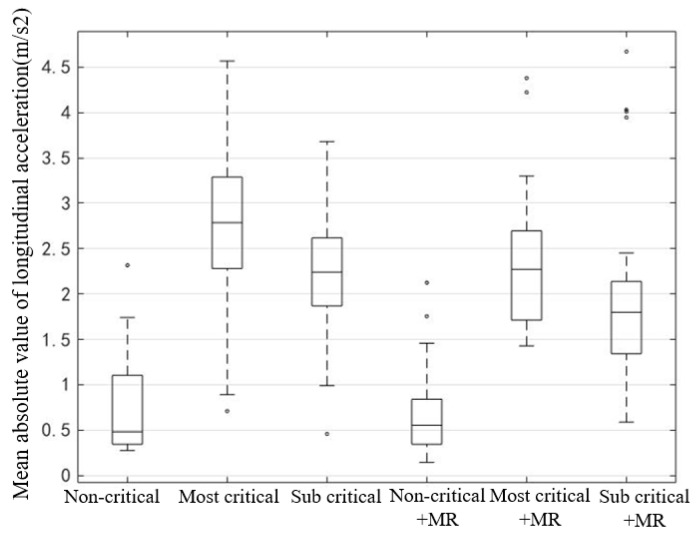
Box plot of the mean absolute value of longitudinal acceleration of the vehicle in 5 s after taking over.

**Figure 17 ijerph-19-00412-f017:**
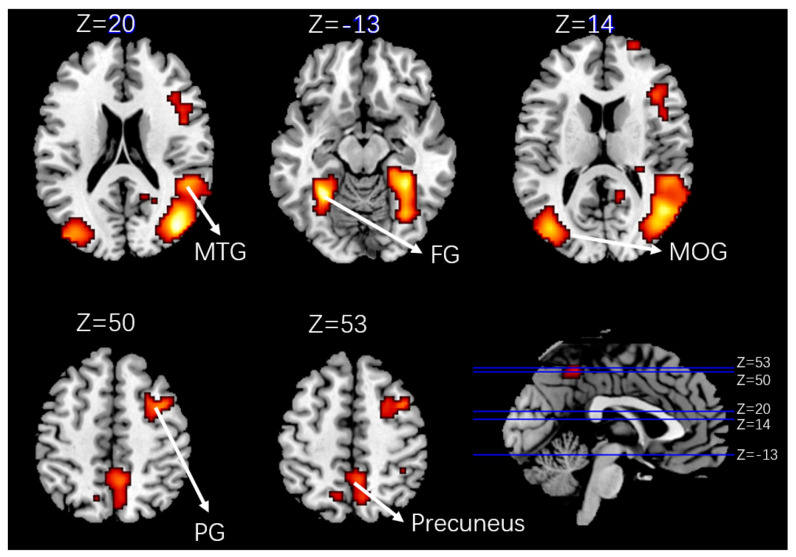
Activation brain area profile (MTG = Middle temporal gyrus; FG = Fusiform gyrus; MOG = Middle occipital gyrus; PG = precentral gyrus).

**Table 1 ijerph-19-00412-t001:** Indicators of six driving scenarios.

Scenario	Duration of Self-Driving before Taking Over	Lead Vehicle Speed (km/h)	Ego Vehicle Speed (km/h)	Reserved Distance	Scenario
Non-critical	210 s	50	50	41.7	No
Most-critical	210 s	50	80	41.7 (5 s to collision)	No
Sub-critical	210 s	50	80	58.3 (7 s to collision)	No
Non-critical + MR	210 s	50	50	41.7	Yes
Most-critical + MR	210 s	50	80	41.7 (5 s to collision)	Yes
Sub-critical + MR	210 s	50	80	58.3 (7 s to collision)	Yes

**Table 2 ijerph-19-00412-t002:** Setting of stimulation scenes.

Scenario	Lead Vehicle Speed (km/h)	Ego Vehicle Speed (km/h)	Reserved Distance (m)
Non-critical	50	50	41.7
Critical	50	80	41.7 (5 s to collision)

**Table 3 ijerph-19-00412-t003:** Scanning process.

	50 s	10 s	50 s	10 s	…
1 (6 min)	FIX	Critical Stimulus	FIX	Non-critical Stimulus	…A total of 6 stimuli
2 (6 min)	FIX	Non-critical Stimulus	FIX	Critical Stimulus	…A total of 6 stimuli
3 (6 min)	FIX	Critical Stimulus	FIX	Non-critical Stimulus	…A total of 6 stimuli

**Table 4 ijerph-19-00412-t004:** Average (AD) and standard deviation (SD) of visual reaction time (seconds).

Scenario	Non-Critical	Most-Critical	Sub-Critical	Non-Critical + MR	Most-Critical + MR	Sub-Critical + MR
AD	2.055	2.045	1.973	0.426	0.320	0.399
SD	0.629	0.699	0.753	0.590	0.437	0.499

**Table 5 ijerph-19-00412-t005:** Total fixation time in Road area ahead (s).

Non-Critical	Most-Critical	Sub-Critical	Non-Critical + MR	Most-Critical + MR	Sub-Critical + MR
5.689	6.4	6.655	7.479	7.857	7.733

**Table 6 ijerph-19-00412-t006:** *T*-test results of the influence of variables on pupil diameter.

Variables	Scenes	AVG	SD	SEM	Confidence Interval of 0.95		Significance
					Lower Limit	Upper Limit	
Speed difference between the lead and ego vehicle	1, 2	−0.144	0.242	0.065	−0.284	−0.004	0.045
	4, 5	−0.102	0.141	0.038	−0.184	−0.021	0.018
Reserved safety distance	2, 3	0.072	0.177	0.047	−0.030	0.174	0.152
	5, 6	0.019	0.165	0.044	−0.076	0.114	0.676
MR	1, 4	0.232	0.384	0.103	0.010	0.454	0.041
	2, 5	0.274	0.346	0.092	0.074	0.473	0.011
	3, 6	0.220	0.416	0.111	−0.020	0.461	0.069

Note: (1) Scene 1 refers to non-critical scenario. (2) Scene 2 refers to most-critical scenario. (3) Scene 3 refers to sub-critical scenario. (4) Scene 4 refers to non-critical with MR scenario. (5) Scene 5 refers to most-critical with MR scenario. (6) Scene 6 refers to sub-critical with MR scenario.

**Table 7 ijerph-19-00412-t007:** Number of people with different takeover modes in six scenarios.

Scenario	Non-Critical	Most-Critical	Sub-Critical	Non-Critical + MR	Most-Critical + MR	Sub-Critical + MR
Mode A	31	31	29	39	39	38
Mode B	8	8	10	0	0	1

**Table 8 ijerph-19-00412-t008:** Paired sample *t*-test of mean speed within 5 s after taking over.

Variables	Scenes	AVG	SD	SEM	Confidence Interval of 0.95		Significance
					Lower Limit	Upper Limit	
Reserved safety distance	2, 3	−7.130	9.521	1.525	−10.216	−4.043	0.000
	5, 6	−5.035	7.910	1.267	−7.599	−2.471	0.000
MR	1, 4	−1.389	6.552	1.049	−3.513	0.735	0.193
	2, 5	−4.031	11.378	1.822	−7.719	−0.342	0.033
	3, 6	−1.936	7.275	1.165	−4.294	0.422	0.105

Note: (1) scene 1 refers to non-critical scenario; (2) scene 2 refers to most-critical scenario; (3) scene 3 refers to sub-critical scenario; (4) scene 4 refers to non-critical with MR scenario; (5) scene 5 refers to most-critical with MR scenario; (6) scene 6 refers to sub-critical with MR scenario.

**Table 9 ijerph-19-00412-t009:** Paired sample *t*-test of absolute value of longitudinal acceleration within 5 s after taking over.

Variables	Scenes	AVG	SD	SEM	Confidence Interval of 0.95		Significance
					Lower Limit	Upper Limit	
Reserved safety distance	2, 3	0.553	0.982	0.157	0.234	0.871	0.001
	5, 6	0.402	0.698	0.112	0.176	0.628	0.001
MR	1, 4	0.060	0.651	0.104	−0.151	0.272	0.566
	2, 5	0.448	1.108	0.177	0.089	0.807	0.016
	3, 6	0.297	0.889	0.142	0.009	0.585	0.044

Note: (1) scene 1 refers to non-critical scenario; (2) scene 2 refers to most-critical scenario; (3) scene 3 refers to sub-critical scenario; (4) scene 4 refers to non-critical with MR scenario; (5) scene 5 refers to most-critical with MR scenario; (6) scene 6 refers to sub-critical with MR scenario.

**Table 10 ijerph-19-00412-t010:** Paired sample *t*-test of standard deviation of lateral displacement within 5 s after taking over.

Variables	Scenes	AVG	SD	SEM	Confidence Interval of 0.95		Significance
					Lower Limit	Upper Limit	
Speed difference between the lead and ego vehicle	1, 2	−0.047	0.115	0.019	−0.085	−0.008	0.018
	4, 5	−0.026	0.079	0.013	−0.053	0.000	0.050
Reserved safety distance	2, 3	0.028	0.132	0.022	−0.016	0.072	0.209
	5, 6	−0.013	0.100	0.016	−0.047	0.020	0.422
MR	1, 4	0.003	0.068	0.011	−0.020	0.025	0.809
	2, 5	0.023	0.118	0.019	−0.016	0.062	0.241
	3, 6	−0.018	0.104	0.017	−0.053	0.017	0.297

Note: (1) scene 1 refers to non-critical scenario; (2) scene 2 refers to most-critical scenario; (3) scene 3 refers to sub-critical scenario; (4) scene 4 refers to non-critical with MR scenario; (5) scene 5 refers to most-critical with MR scenario; (6) scene 6 refers to sub-critical with MR scenario.

**Table 11 ijerph-19-00412-t011:** Number of people with different takeover modes in six scenarios.

Number of Active Voxels	*t*-Score	*p*-Value	Standard Coordinates (x,y,z) (mm)	Bradman’s Area	AAL Area
1302	7.71	<0.001	42, −73, 20	BA39	Occipital_mid_R
	6.77		45, −64, 17		Temporal_mid_R
167	7.03	0.004	−30, −49, −13	BA37	Fusiform_L
531	5.91	<0.001	−39, −76, 14	BA19	Occipital_mid_L
160	5.06	0.005	36, 5, 50	BA6	Precentral_R
146	4.21	0.007	9, −64, 53	BA7	Precuneus_R

Note: (1) L represents the left brain and R represents the right brain; (2) BA39: angular gyrus, part of Wernicke’s area; BA37: fusiform gyrus; BA19: associative visual cortex; BA6: pre-motor and supplementary motor cortex; BA7: somatosensory association cortex.

**Table 12 ijerph-19-00412-t012:** Voxel number, *t*-score, coordinates and deactivated brain location in deactivation area.

Number of Deactive Voxels	*t*-Score	*p*-Value	Standard Coordinates (x,y,z) (mm)	Bradman’s Area	AAL Area
35	3.98	0.144	−42, −19, 65	4	Precentral_L

Note: L represents the left brain.

## Data Availability

The data that support the findings of this study are available from the corresponding author, upon reasonable request.

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
