# Peer review of "How Does Approaching a Lead Vehicle and Monitoring Request Affect Drivers’ Takeover Performance? A Simulated Driving Study with Functional MRI"

_ijerph, 2021, doi:10.3390/ijerph19010412_

Round 1

Reviewer 1 Report

This paper describes two types of interesting experiments. In general the paper is of a good quality.

Two types of experiments were used. It seems that the participants of the driving simulation experiment and the participants of the fMRI experiment are not the same persons. Please explain why the participants of both experiments are not the same persons. Does this have any influence on your results?

Section 5, conclusions, only describes the conclusions from your own experiments. However, there should also be a discussion regarding the results found by other researchers and your own results.

Reviewer 2 Report

The paper is a well-done study of driver takeover of vehicle control in the context of the automated driver. The results are well substantiated and analyzed.

However, the (driving) simulator setup is very simple, and it is at least questionable whether this simple setup can accurately enough reflect the actual behavior of subjects in real road traffic. This is a bit of a pity, since on the other hand the use of the fMRI in the further investigations involved a great deal of effort in monitoring the test subjects.

The explanation of the fMRI experiment in lines 175-182 should be clarified again. How were the "experimental variables" simplified? What exactly was done becomes clear in the following text sections. Nevertheless, I recommend a clarification.

Why is the time for the monitoring request just set at 30 seconds? Have there been any preliminary studies on this?

Explanation of how the formula (1) is obtained. How are the individual variables specified. For example, what does "x_head movement 2" mean?

It would be interesting to find out whether a haptic takeover prompt does not produce better results than a purely visual one.

Zeile 299. Schreibfehler: „potin“ muss heißen „point“

Line 300: Missing space between "dashboard" and "(Figure 8)".

Line 334: Do the authors have any clues as to where the difference in pupil diameters between left and right eyes came from. Does this have to do with right-hand traffic?

Line 400: Check the characters in the "Scenes" column.

Likewise, the other tables, e.g., line 429, 444, etc.
